

# Does plastic type matter? Insights into non-indigenous marine larvae recruitment under controlled conditions

François Audrézet[1,2], Anastasija Zaiko[1,2], Patrick Cahill[1], Olivier Champeau[1], Louis A. Tremblay[1,3], Dawn Smith[4], Susanna A. Wood[1], Gavin Lear[3] and Xavier Pochon[1,2]

[1] Cawthron Institute, Nelson, New Zealand
[2] University of Auckland, Institute of Marine Science, Auckland, New Zealand
[3] University of Auckland, School of Biological Sciences, Auckland, New Zealand
[4] Rua Bioscience, Ruatorea, New Zealand

Corresponding author
François Audrézet,
audrezet.francois@gmail.com

## ABSTRACT

Marine plastic debris (MPD) are a global threat to marine ecosystems. Among countless ecosystem impacts, MPD can serve as a vector for marine 'hitchhikers' by facilitating transport and subsequent spread of unwanted pests and pathogens. The transport and spread of these non-indigenous species (NIS) can have substantial impacts on native biodiversity, ecosystem services/functions and hence, important economic consequences. Over the past decade, increasing research interest has been directed towards the characterization of biological communities colonizing plastic debris, the so called Plastisphere. Despite remarkable advances in this field, little is known regarding the recruitment patterns of NIS larvae and propagules on MPD, and the factors influencing these patterns. To address this knowledge gap, we used custom-made bioassay chambers and ran four consecutive bioassays to compare the settlement patterns of four distinct model biofouling organisms' larvae, including the three notorious invaders *Crassostrea gigas*, *Ciona savignyi* and *Mytilus galloprovincialis*, along with one sessile macro-invertebrate *Spirobranchus cariniferus*, on three different types of polymers, namely Low-Linear Density Polyethylene (LLDPE), Polylactic Acid (PLA), Nylon-6, and a glass control. Control bioassay chambers were included to investigate the microbial community composition colonizing the different substrates using 16S rRNA metabarcoding. We observed species-specific settlement patterns, with larvae aggregating on different locations on the substrates. Furthermore, our results revealed that *C. savignyi* and *S. cariniferus* generally favoured Nylon and PLA, whereas no specific preferences were observed for *C. gigas* and *M. galloprovincialis*. We did not detect significant differences in bacterial community composition between the tested substrates. Taken together, our results highlight the complexity of interactions between NIS larvae and plastic polymers. We conclude that several factors and their potential interactions influenced the results of this investigation, including: (i) species-specific larval biological traits and ecology; (ii) physical and chemical composition of the substrates; and (iii) biological cues emitted by bacterial biofilm and the level of chemosensitivity of the different NIS larvae. To mitigate the biosecurity risks associated with drifting plastic debris, additional research effort is critical to effectively decipher the mechanisms involved in the recruitment of NIS on MPD.

## INTRODUCTION

Plastic pollution in natural ecosystems has become one of the major environmental issues of the twenty-first century (*Galgani, Pham & Reisser, 2017*). Since mass production of petrochemical-derived polymers began in the 1950s, humanity has produced a staggering amount and diversity of plastic materials with an estimated global annual production of 330 million metric tons (Mt) in 2016 (*PlasticsEurope, 2021*). In 2017, *Geyer, Jambeck & Law (2017)* estimated that approximately 6,300 Mt of plastic waste had been generated, of which 9% had been recycled, 12% was incinerated, and 79% had accumulated in landfills or in the natural environment. Ironically, the same physical properties (*i.e.*, durability, lightweight, malleability, low processing cost) that have made plastic so commercially successful are now creating unprecedented environmental concerns across terrestrial, freshwater and marine ecosystems (*Boucher & Billard, 2019*). Today, the marine environment is the main hub of mismanaged plastic waste, with an estimated 8.4 Mt of plastic waste entering the world's oceans every year (*Jambeck et al., 2015*). For example, the 2016 US plastic waste inputs to the coastal environment were among the highest in the world, representing between 0.51 to 1.45 Mt (*Law et al., 2020*).

The impacts of plastic debris on marine biota have been extensively described. They include ingestion (*Santos, Machovsky-Capuska & Andrades, 2021*), entanglement (*Jepsen & de Bruyn, 2019*), and other potential biological impacts through food web interference and release of toxic compounds (*Teuten et al., 2009*; *Setälä, Fleming-Lehtinen & Lehtiniemi, 2014*). Recent studies have highlighted emerging impacts on species biodiversity and biogeography, with marine plastic debris (MPD) acting as effective vectors for the transport of unwanted organisms including non-indigenous species (NIS) and pathogens from coastal to open ocean environments (*Audrézet et al., 2020*; *Haram et al., 2021*). Marine plastic debris provide a long-lived and very common submerged surfaces on which micro- and macro-colonizing species thrive and are dispersed to new locations (*Barnes & Milner, 2005*; *Carlton et al., 2017*). In 2013, *Zettler, Mincer & Amaral-Zettler (2013)* coined the term "Plastisphere" to characterize the diverse microbial assemblages of organisms attached to plastic surfaces. This pioneering publication triggered numerous investigators to characterize microbial communities inhabiting the plastisphere, including bacteria (*Zettler, Mincer & Amaral-Zettler, 2013*; *Frère et al., 2018*), fungi (*Lacerda et al., 2020*), diatoms (*Cheng et al., 2021*), putative pathogens (*Kirstein et al., 2016*; *Viršek et al., 2017*), and potential plastic degraders (*Erni-Cassola et al., 2020*; *Wallbank et al., 2022*). However, despite remarkable advances in characterizing the micro-plastisphere on various polymer types, little is known regarding the mechanisms involved in macro-plastisphere community succession and the factors influencing the recruitment of macro-invertebrates, especially NIS larvae and propagules.

In this study, we ran four consecutive microcosm experiments using custom made bioassay chambers (*Pansch et al., 2017*) to compare the larval settlement strategies of four model macrofouling invertebrates, including three notorious invaders: the Pacific oyster *Crassostrea gigas*, the Pacific transparent sea squirt *Ciona savignyi*, the blue mussel *Mytilus galloprovincialis*, and the blue tubeworm *Spirobranchus cariniferus*. Recruitment was assessed on three different polymer types (low-linear density polyethylene–LLDPE; Nylon-6; and polylactic acid–PLA), and a glass control. Polymers were selected for their prevalence in marine ecosystems, and their specific physical properties (*i.e.*, LLDPE is a low surface energy (LSE) polymer, whereas PLA, nylon and glass are high surface energy (HSE) substrates). *Rittschof et al. (1998)* demonstrated that invertebrate larvae can sense surface energy and adapt to select an optimal substrate for settlement. Hence, polymers were selected based on these properties, to investigate if surface energy had an influence on marine invertebrates' recruitment. The aim of this study was to investigate whether the larvae of macrofouling NIS would exhibit preferences for a particular substrate type in controlled conditions. In parallel, control bioassay chambers were used to characterize bacterial communities' composition at the end of each microcosm experiment using metabarcoding analysis. We hypothesized that; (i) settlement strategies and affinity for specific substrate types would vary among the macrofouling species and this would be related to species-specific ecological traits; and (ii) differences in larval recruitment among substrates would be affected by bacterial biofilm community composition.

## METHODS

### Macrofouling species, larval spawning, and culturing

*Crassostrea gigas* larvae were cultured in a hatchery under controlled conditions (*Rico-Villa, Pouvreau & Robert, 2009*, *Vignier et al., 2021*). Briefly, adult oysters were transferred to the Cawthron Aquaculture Park (CAP; Nelson New Zealand) hatchery for conditioning and fed *ad libitum* with bulk cultured *Isochrysis galbana* ($8–9 \times 10^6$ cells ml$^{-1}$) and *Pavlova lutheri* ($10–12 \times 10^6$ cells ml$^{-1}$). Fully mature oysters were strip-spawned according to *Allen & Bushek (1992)* and gametes were collected and fertilized. Embryos were then incubated in static 170-L tanks at 23 °C for 24 h, and D-larvae were transferred to 170-L flow-through rearing systems and continuously fed with a mixed diet of *Chaetoceros calcitrans* (CS-178) and *Tisochrysis lutea* (CS-177) throughout rearing. After 17 days, larvae developed into the pediveliger stage and were competent to settle.

*Ciona savignyi* adults collected from the underside of pontoons at the Nelson Marina (Nelson, New-Zealand–Lat: 41°16′ 14.81″ S; Long: 173° 17′ 2.54″ E) were housed in water lily baskets suspended in 50-L glass aquaria for up to 3 weeks. Aquaria were held at 18 ± 1 °C (mean ± standard error), 34 ± 1 Practical Salinity Unit (psu). Constant full-spectrum fluorescent light prevented premature spawning. Every day the filtration of the aquaria was paused for 3 h while *C. savignyi* were fed 250 mL of an $8–9 \times 10^6$ cells mL$^{-1}$ *Isochrysis galbana* culture. Three gravid individuals with densely packed egg and sperm ducts were spawned according to *Cahill et al. (2016)*. Following spawning, larvae were transferred to conical flasks and diluted with reconstituted seawater (RSW; 33 ± 0.5 psu; Red Sea Salt, Red Sea Aquatics, Cheddar, UK) to yield desired larval densities.
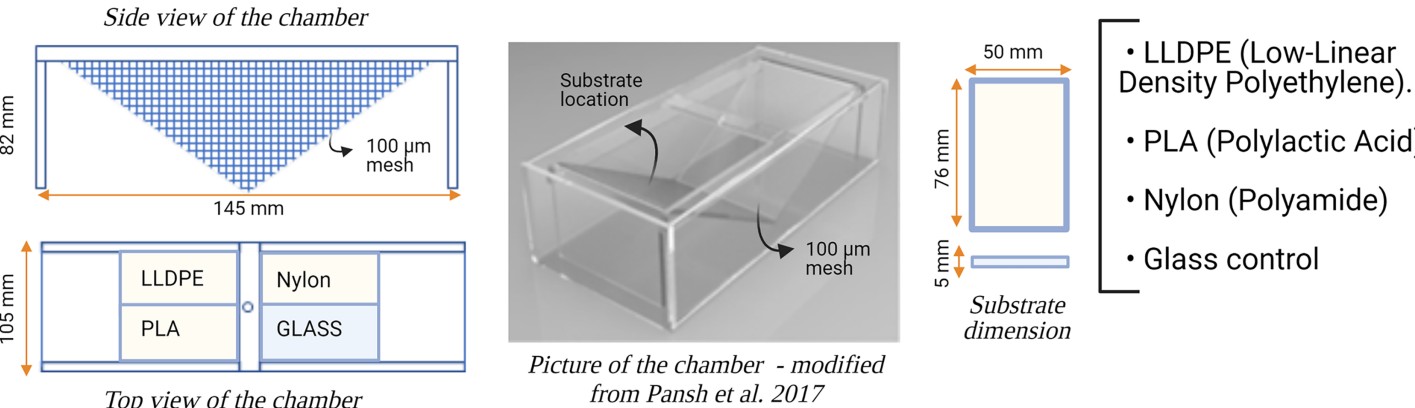

**Figure 1** **Design of the bioassay chamber and the substrates selected to compare the settlement of invasive species larvae.** A 100 μm mesh size was added on each side of the chambers to prevent the larvae from escaping.

*Spirobranchus caraniferus* adults were collected from Delaware Bay (Nelson, New-Zealand – Lat: 41°09′ 33.6″ S; Long: 173° 28′ 34.1″ E). After collection, spawning was induced by removing the external calcareous tube of the worms and placing them together in a 100-mL glass beaker filled with RSW as described by *Brooke et al. (2018)*. The eggs and sperm were left for 1 h to fertilize and then transferred to a 10-L conical flask filled with UV-sterilized filtered (0.4 μm) seawater (FSW; temperature 18 ± 1 °C; salinity 34 ± 1 ppt; pH 8.0 ± 0.2) and aerated with an aquarium bubbler. Larvae were cultured for 15 days with daily replacement of FSW and fed with a bispecific diet of *I. galbana* as per *C. savignyi*. The 15-day-old larvae were made competent to settle by exposure to $10^{-3}$ M 3-isobutyl-1-methylxanthine for 4 h and were then rinsed five times with RSW before being placed in the bioassay chambers.

*Mytilus galloprovincialis* adults were collected from submerged ropes at Elaine Bay in the Marlborough Sounds (Lat: 41° 3′ 19″ S; Long: 173° 46′ 9″ E, New Zealand). Mature individuals were induced to spawn by thermal stimulation, as described by *His, Seaman & Beiras (1997)*. After spawning, fertilization and embryo-larval development were carried out according to (*ASTM International, 2021*). Larvae were kept in conical flasks, each containing 5-L of FSW and one aquarium bubbler stone to promote gentle mixing. Larvae were kept in culture and fed with a mixed diet of *C. calcitrans* (CS-178) and *T. lutea* (CS-177), with a daily change of FSW. After 23 days, larvae developed to the pediveliger stage and were competent for settlement.

## Bioassay chambers, polymer production and study design

Six flow-through bioassay chambers were fabricated by Nelson Plastic Ltd. (Nelson, New-Zealand) following a design modified from *Pansch et al. (2017)*. Each chamber consists of an inner V-shaped stand in which four panels (*i.e.*, three different types of polymer tokens and a glass control) are placed at a 120° angle facing each other (Fig. 1). Six bioassay chambers (145 mm [L] × 105 mm [W] × 82 [H]) were used during each consecutive microcosm experiment.

Three different types of polymer tokens (50 mm × 76 mm), manufactured by Scion (Rotorua, New Zealand), were used as the 'Substrate' treatment in each microcosm experiment (Fig. 1). The tokens were injection-moulded from formulations of each investigated plastic polymer as follows: LLDPE–base LLDPE with Irganox 1076 (CAS 2082-79-3) and 0.25% Irganox B215 (Irganox B215 = 67%) Irgafox® 168 (CAS 31570-04-4) and 33% Irganox® 1010 (CAS 6683-19-8); PLA–Ingeo 3052D–blended with ethylene bis (stearamide) (CAS 110-30-5); Nylon-6–Ultramid B3S with talc and 0.5% Nylostab S-EED (CAS 42774-15-2). The polymer types used in this study (LLDPE, PLA and Nylon-6) contained additives typically included in the manufacture of these products for UV light stability and degradation prevention.

Four consecutive bioassay microcosm trials were undertaken between December 2020 and March 2021. The first bioassay was performed with *C. gigas* larvae (A1, 7-16 December 2020) followed by *C. savignyi* (A2, 11-20 January 2021), *S. caraniferus* (A3, 19-28 February 2021) and *M. galloprovincialis* (A4, 15-24 March 2021). During each trial (*i.e.*, for each species), six bioassay chambers were supplied with 2.5 L of filtered (0.2 µm), seawater h$^{-1}$ as part of a 1,000-L recirculating system held at 18 ± 1 °C, and 34 ± 1 psu for the duration of the experiment. The seawater was not replaced between assays. Filtration of the recirculating seawater system consisted of a Dacron screen and a trickling biofilm filter filled with ~0.125 m$^3$ of generic 'bioball' filter media. Bioassay chambers were kept under a 12:12 light to dark regime.

Each experiment consisted of: (i) Control assay: three bioassay chambers were kept in the recirculating system with constant seawater supply for 7 days to allow biofilm growth; (ii) Settlement assay: the remaining three bioassay chambers were kept in the same recirculating system with constant seawater supply for 7 days. After 7 days, 500 competent larvae of the respective model macrofouling species were placed in each of the three settlement assay aquaria for 48 h to investigate preferential settlement on each substrate (Fig. 2).

After 9 days (7 days of biofilm formation and 2 days of larval settlement) polymer and glass tokens were collected individually. For each experiment, there were 24 samples in total (six samples of LLDPE, PLA, Nylon-6 and glass controls). The 12 tokens from the settlement assays were used for microscopic analysis of larval settlement and were kept in a glass container with RSW for up to 1 h. The 12 tokens from the control assays were used for bacterial characterization were individually placed in Fisherbrand™ sterile sampling bags and kept on ice until processing within 2 h (Fig. 3).

## Larvae counts and visualization of settlement location

Counts of larval settlement were performed with a dissecting microscope at 20× magnification (RS PRO Microscope, Norman King, Beauvais Cedex). The total number of settled larvae was recorded in each instance, along with specific location of larvae on the tokens as a proxy for larval aggregation. The location of the larvae on the different substrates was assessed visually with a dissecting microscope.

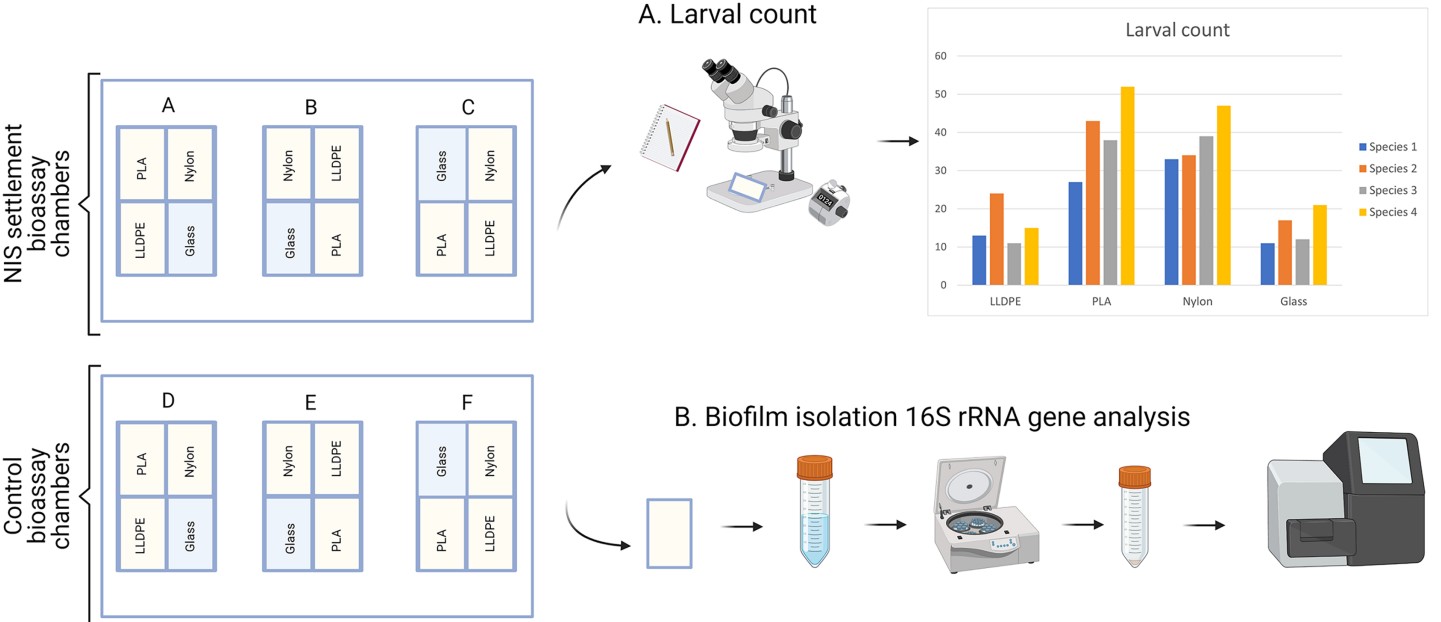

**Figure 2 Invasive species settlement assay.** The invasive species settlement assay consists of 7 days of biofilm development followed by 2 days of larval settlement in controlled conditions. Each step was performed four times, once for each of the four organisms of study. PLA = Polylactic Acid, LLDPE = Low-Linear Density Polyethylene.

**Figure 3 Sample processing for non-indigenous species (NIS) settlement bioassay and control assay.** For NIS settlement bioassay, larval count and location were conducted directly after sample collection (A). For the control assay, biofilm isolation for metabarcoding analysis of bacterial communities was conducted directly after sample collection (B). PLA = Polylactic Acid, LLDPE = Low-Linear Density Polyethylene.

## Metabarcoding of bacterial communities

Each step of the following molecular analysis (Fig. 3) was conducted in separate sterile laboratories with sequential workflow to eliminate cross-contamination. Rooms dedicated to DNA extraction, amplification set-up and template addition were equipped with

laminar flow cabinets with HEPA filtration and room-wide ultra-violet sterilization which was switched on for >15 min before and after each use (*Pearman et al., 2020*). Aerosol's barrier tips (Axygen, San Francisco, CA, USA) were used throughout.

Within 2 h, the sterile sampling bags containing the polymer tokens from the control assays were filled with 30 mL of ice-cold Tris-EDTA Buffer solution (Tris 1 mM, EDTA 1 mM; prepared from sterile, Ultrapure water, Ultrapure Tris pH 8.0 and Ultrapure EDTA pH 8.0) and sonicated for 2 min at 50 Hz in an ice-cold ultrasonic water bath to recover the attached biofilm fraction (Bandelin Sonorex RF 100H, 50–60 Hz, Sigma-Aldrich, USA). Following sonication, each homogenate was poured into separate sterile 50 mL Falcon tubes (Cat No. 227–261, Greiner Cellstar®, Sigma-Aldrich New Zealand). The sonicate solution was centrifuged ($4,500 \times g$, 10 min, 4 °C). Supernatants were gently decanted and discarded, followed by an additional 5 min centrifugation step and removal of the remaining supernatant with a pipette (*Wallbank et al., 2022*).

Microbial DNA was extracted individually from each pelleted biofilm sample using the PowerSoil® DNA Isolation Kit (QIAGEN, MOBIO, Carlsbad, USA) following the manufacturer's instructions. DNA was extracted from a total of 60 samples which was comprised of 12 samples for each of the bioassay trials, plus three procedural control samples per bioassay trial (one TAE buffer control, one seawater control before adding the larvae, one seawater control after adding the invasive larvae), along with extraction kit control blanks. Each sample was eluted in a final volume of 50 μL of elution buffer.

The V3-V4 regions of the bacterial 16S ribosomal RNA (16S rRNA) gene was amplified by Polymerase Chain Reaction (PCR), using the bacterial specific primers 341F: 5′-CCT ACG GGN GGC WGC AG-3′ and 805R: 5′-GAC TAC HVG GGT ATC TAA TCC-3′ (*Herlemann et al., 2011*; *Klindworth et al., 2013*). Both primer sets contained an Illumina overhang adapter for NEXTERA indexing, as described by *Pochon et al. (2019)*. PCR reactions were undertaken in an Eppendorf Mastercycler (Eppendorf, Hamburg, Germany) in a total volume of 50 μL using MyFiTM PCR Master Mix (Bioline Meridian Bioscience, Memphis, Tennessee, USA), including 2 μL of each primer (10 mM stock) and 2 μL of template DNA. The PCR cycles for the 16S rRNA gene amplification were as follows: 94 °C for 3 min followed by 35 cycles of 94 °C (20 s), 52 °C (20 s) and 72 °C (30 s) with a final extension at 72 °C for 5 min. Negative (no-template) PCR controls were included in each PCR run (*Audrézet et al., 2022*). Amplicon PCR products were purified using AMPure XP PCR Purification beads (Agencourt, Beverly, MA, USA), quantified using a Qubit Fluorometer (Life Technologies, Carlsbad, CA, USA) and diluted to 3 ng μL$^{-1}$. An additional water control was added to test for potential contamination during the sequencing workflow. Normalized PCR products and controls ($n$ = 63; 54 samples, five extraction blanks, and four PCR blanks) were sent for library preparation and sequencing on an Illumina MiseqTM platform at Auckland Genomics, University of Auckland, New Zealand following the Illumina16S rRNA metagenomics library preparation manual (*D'Amore et al., 2016*). Sequencing adapters and sample-specific indices were added to each amplicon *via* a second round of PCR using a Nextera Index kit. After that, 5 μL of each indexed sample was pooled, and a single clean-up of pooled PCR products was undertaken, as previously described by *Audrézet et al. (2022)*. A bioanalyzer
was used to check the quality of the library which was then diluted to 4 nM and denatured. The library was diluted to a final loading concentration of 7 ρM with a 15% spike of PhiX. Paired-end sequences (2 bp × 250 bp) were generated on an Illumina MiSeq instrument. Raw sequences were deposited in the NCBI short read archive under accession: PRJNA836386.

## Bioinformatics and statistical analyses

For larval settlement data, one-way analysis of variance (ANOVA) was performed on the larval count results for the factor 'Substrate' using the 'Vegan' package in RStudio (*Oksanen et al., 2013*; *R Development Core Team, 2013*). Differences at $p \leq 0.05$ were deemed statistically significant. Following ANOVA analysis, Tukey's honestly significant difference test (Tukey's HSD) was performed in RStudio to test differences between substrate types with the R package *agricolae* (*De Mendiburu, 2014*).

For metabarcoding data, raw sequence reads (with Illumina adapter sequences removed by the sequencing instrument) were trimmed using cutadapt v2.10 to remove primer DNA sequences (*Martin, 2011*), with no primer mismatch allowed. Quality filtering, denoising, merging pair-end sequences, and calling amplicon sequence variants (ASVs) were performed using the DADA2 version 1.20.0 package, implemented in R version 4.0.5 (*Callahan et al., 2016*). Following exploration of the DNA sequence quality plots, sequences were trimmed at a length of 220 for both forward and reverse reads, two or six errors were allowed for forward or reverse reads respectively. Reads were truncated at a quality score less than 2, and the maximum number of ambiguous nucleotides was set to zero. Singleton ASVs data (*i.e.*, isolated sequences that were observed only once in the dataset) were removed to overcome sequencing errors (*Tedersoo et al., 2010*, *Caporaso et al., 2011*, *Edgar, 2013*). The remaining paired-end reads were merged with a minimum overlap of 25 bp and one mismatch allowed in the overlap region. Chimaera removal was performed using the default (consensus) method and the de-noised ASVs taxonomically classified against the SILVA 132 database for 16S rRNA (*Quast et al., 2012*) using DADA2 "assign Taxonomy" command, based on the RDP classifier (*Wang et al., 2007*).

The 16S rRNA dataset was filtered to exclude any ASVs classified as Eukaryota in the rank Kingdom, Chloroplast in the rank Class, and Mitochondria in the rank Family using the "subset_taxa" command implemented within the R package phyloseq (*McMurdie & Holmes, 2013*). The maximum number of ASVs found across negative controls was subtracted from the corresponding ASVs to offset potential contamination noise (*Bell et al., 2019*; *Clark et al., 2020*). Rarefaction curves were plotted using the '*ggrare*' function in R (package *ranacapa*; *Kandlikar et al. (2018)*). The 16S rRNA rarefaction curves indicated that the sequencing depth attained per sample adequately captured biodiversity (*i.e.*, the curves have reached a plateau). However, 10 samples yielded an extremely low post-filtering number of reads (<2,000) and were removed from further analysis (Table S1).

The community structure analyses were performed on the unrarefied dataset transformed into proportional read abundance. An ASV table generated by the bioinformatic pipeline was uploaded into the Plymouth Routines in Multivariate Ecological Research (PRIMER 7) v7.0.13 software (*Anderson, 2001*; *Clarke & Gorley,*
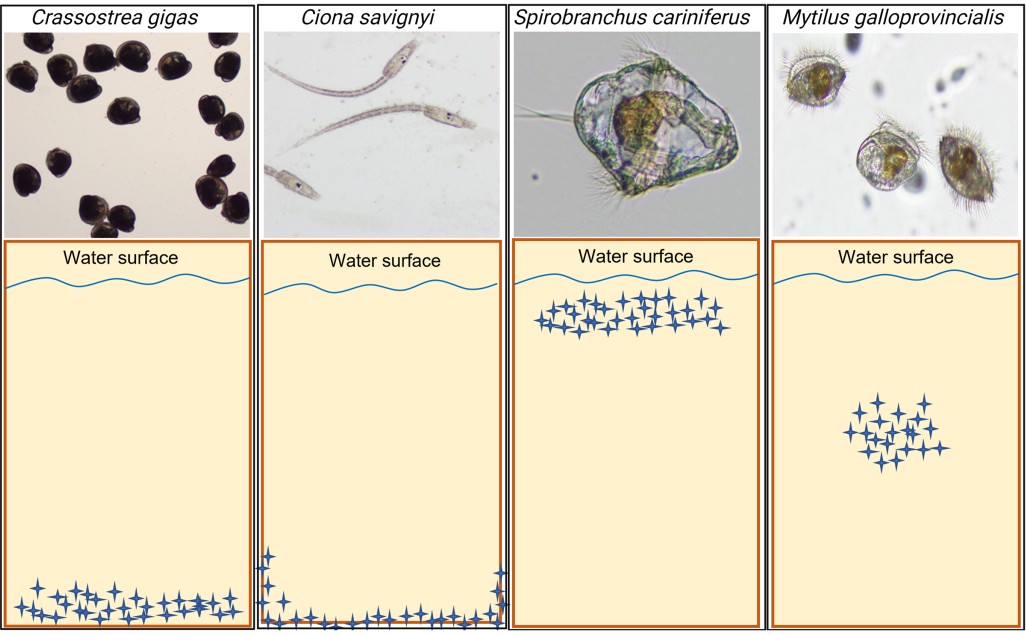

**Figure 4 Schematic visualization of the observed non-indigenous species larval settlement patterns. Blue stars represent the larvae visualized during the microscopy investigation.** The larval settlement patterns were quantified visually, in parallel of the larval counting with a dissecting microscope. Due to logistical reasons (time to process the samples rapidly), we could not provide high-definition pictures of the larvae on the substrates' surface. For more information regarding the oysters' larval aggregation, see Figure S1.

*2015*). Square root transformed data was used to construct Bray-Curtis similarity matrix (at ASV level) (*Bukin et al., 2019*), which was used to analyse bacterial community structure for the experimental factors 'Substrate' and 'Assay' (*i.e.*, different polymer types and glass control, and temporal evolution of the bacterial community structure at the end of each microcosm experiment) with PERMANOVA (Permutational Multivariate Analysis of Variance). Before that, Levene's and Shapiro-Wilk's tests were performed to confirm that assumptions for normality and heterogeneity are met. The relative abundance of the 10 most abundant bacterial families related to the different assays (A1-A4) and between substrates was visualized using bar plots generated with the *phyloseq* and *ggplot2* packages in Rstudio. In parallel, differences in bacterial community composition between substrates and among assays were visualized through a Principal Coordinate Analysis (PCoA), based on weighted unifrac distances.

# RESULTS

## Substrate-specific larvae recruitment

The model macrofouling organisms displayed different settlement patterns on each tested substrate (Fig. 4). *Crassostrea gigas* and *C. savignyi* preferentially settled on the lower part of the tested substrates (away from the water surface), with larvae clustering together (Figure S1). In contrast, *S. cariniferus* larvae were densely aggregated in the upper part of

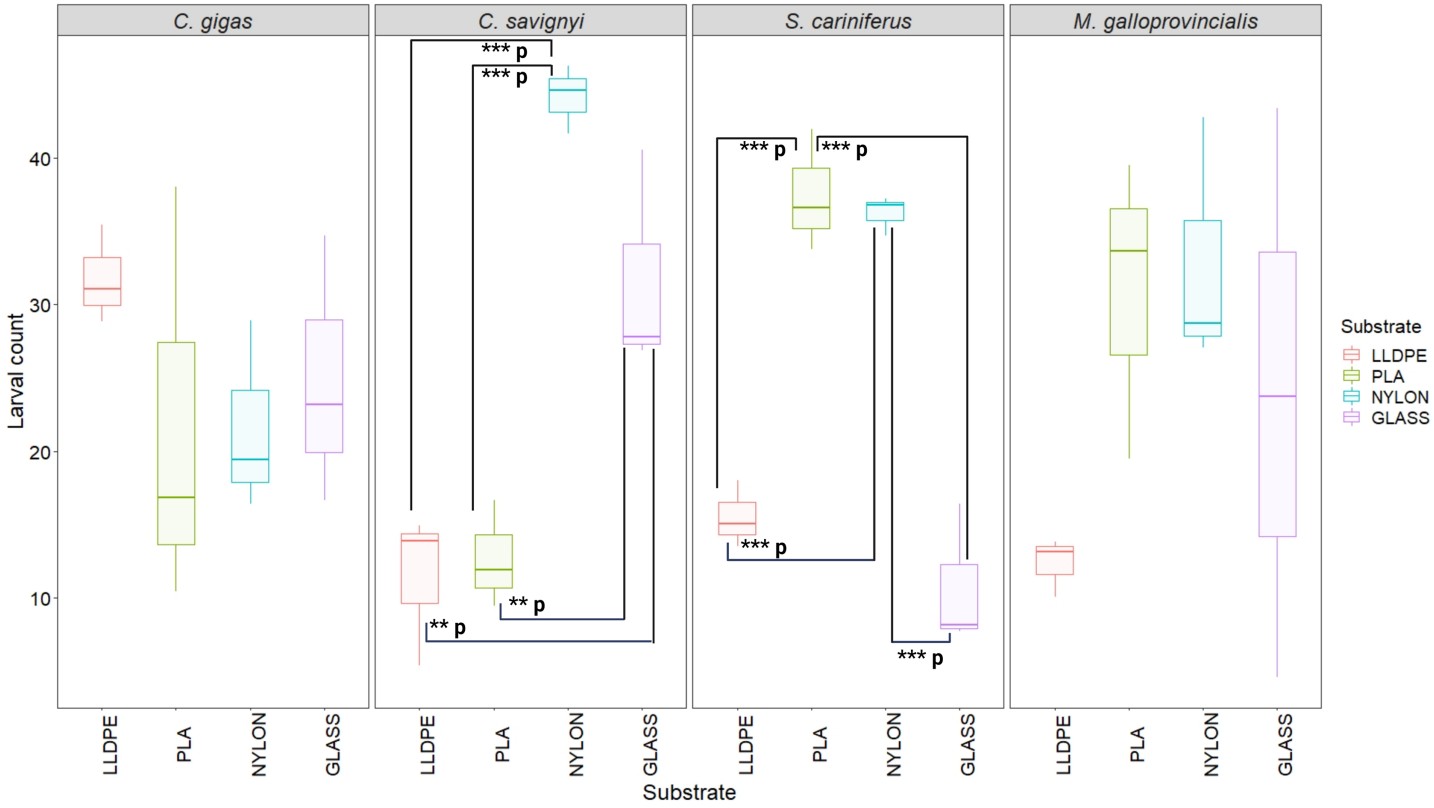

**Figure 5 Boxplot of percentage of larval settlement between substrates and invasive species larvae.** The boxes denote interquartile range (IQR), with the median represented with a line and whiskers extending the most data extreme points. Significant $p$-values are highlighted in bold; $***p = p \leq$ 0.001; $**p = p \leq 0.01$. PLA = Polylactic Acid, LLDPE = Low-Linear Density Polyethylene. *C. gigas = Crassostrea gigas*; *C. savignyi = Ciona savignyi*; *S. cariniferus = Spirobranchus cariniferus*; *M. galloprovincialis = Mytilus galloprovincialis*.

the substrates (near the water surface) (Fig. 4). The pediveligers of *M. galloprovincialis* clustered together on all substrates, aggregating in the center of the tokens (Fig. 4).

The number of settled *C. savignyi* larvae differed significantly among tested substrates (One-way ANOVA $p \leq 0.001$), with maximum recruitment observed on nylon (44 ± 2 larvae per nylon token), and minimum recruitment on LLDPE and PLA (11 ± 5, and 13 ± 3 larvae per token, respectively). A similar pattern was observed for *S. caraniferus*, with significant differences among substrates (One-way ANOVA $p \leq 0.001$). The maximum settlement was associated with PLA and Nylon (38 ± 4 larvae, and 36 ± 2 larvae per token, respectively) and minimum settlement on Glass and LLDPE (11 ± 5, and 16 ± 3 larvae per token, respectively). No preferential recruitment was detected for the two bivalve species: *M. galloprovincialis* ($p = 0.219$) and *C. gigas* ($p = 0.534$) (Fig. 5).

## Biofilm community composition on polymers

There were no significant differences in the bacterial community composition between the four polymer types for each of the sequential control assays (A1-A4), (PERMANOVA, $p = 0.17$; Fig. 6, Figure S2). However, there was a progressive shift in the composition of the bacterial communities over the four sampling periods (Fig. 6), as revealed by the PERMANOVA analysis ($p = 0.001$). Biofilm communities went from being ostensibly

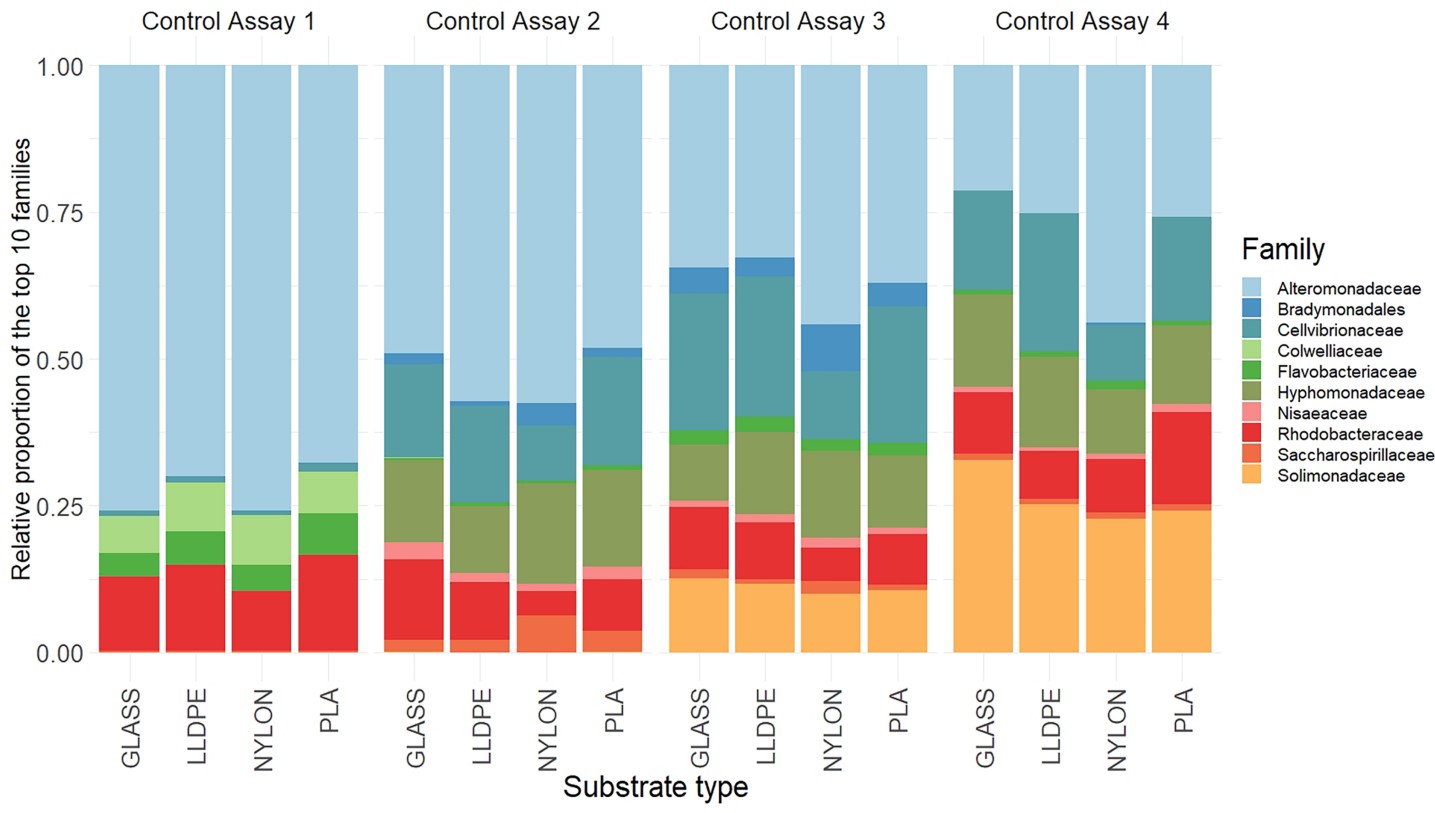

**Figure 6 Relative read abundance of the 10 most dominant bacterial families.** Relative read abundance of the 10 most dominant bacterial families detected on the different substrates in the control assay over four sequential assays (Assay 1 to 4) PLA = Polylactic Acid, LLDPE = Low-Linear Density Polyethylene.

dominated by Alteromonadaceae in Assay 1, to becoming increasingly diverse so that by Assay 4 there were roughly equal proportions of Alteromonadaceae, Cellvibrionaceae, Solimonadaceae, Hyphomonadaceae, and Rhodobacteraceae.

Although the differences within assays were insignificant, some trends between substrate types were observed. During the first assay there was a higher contribution of Alteromonadaceae on nylon and glass (75% and 80%, respectively), compared with LLDPE and PLA (70% and 65%, respectively). In contrast, the relative contribution of Flavobacteriaceae was higher on LLDPE and PLA (17% and 18%, respectively), compared with nylon and glass (12% and 10% respectively). In the second assay, differences in bacterial communities' abundance were mainly associated with Nylon, where the relative contribution of Cellvibrionaceae, and Nisaeaceae were lower than on any other substrates, and Saccharospirillaceae' relative contribution was two times greater than on LLDPE, and glass control (Fig. 6). A similar observation was made during assay 3, where the relative contribution of Bradymonadales was at least twice as high on Nylon compared to any other substrates. In addition, differences in bacterial family contribution were observed on LLDPE in A3. Hyphomonadaceae contributed 20% of the overall diversity on LLDPE, whereas its abundance was on average at 11% on the other substrates. Cellvibrionaceae abundance was lower in comparison with other substrates (12%, 24%, 21%, and 22% on

LLDPE, PLA, nylon, and glass, respectively). The bacterial community was more homogeneous in the final assay (A4). Alteromonadaceae, Cellvibrionaceae, Solimonadaceae, Hyphomonadaceae, and Rhodobacteraceae contributed to more than 90% of the overall bacterial diversity across polymer types and glass control.

## DISCUSSION

We hypothesized that the recruitment of NIS larvae would differ between substrate type (*i.e.* LLDPE, PLA, Nylon, or Glass). Our results supported this hypothesis, with significant differences in larval settlement preferences detected. These trends were species-specific, *C. gigas* and *M. galloprovincialis* settled consistently irrespective of polymer type whereas *C. savignyi* preferred Nylon and *S. caraniferus* preferred PLA and Nylon. These differences likely reflect species-specific larval biological traits and ecology (*Ceccherelli & Rossi, 1984*; *Harris, 2008*; *Gosselin & Sewell, 2013*; *Cahill et al., 2016*), and the physical and chemical properties of the substrates (*Siddik et al., 2019*; *Bae et al., 2022*). Moreover, previous investigators reported that chemical cues released from the bacterial biofilm can affect the recruitment of macro-invertebrates larvae (*Wieczorek & Todd, 1998*; *Hadfield, 2011*). However, given the fact that bacterial community composition was similar across substrates and progressively evolved among assays, we can only speculate that bacterial community composition had an influence on NIS larvae' recruitment.

Our observations of larval recruitment dynamics of *C. gigas*, *S. cariniferus*, *M. galloprovincialis* and *C. savignyi* demonstrated specific larval aggregation on different locations of the tested substrates. For bivalves (*e.g. C. gigas* and *M. galloprovincialis*), serpulid species (*S. cariniferus*) and ascidians (*C. savignyi*), this specific settlement strategy under natural conditions is mainly determined through abiotic stressors such as wave and UV exposure (*Bertness et al., 1999*; *Shafer, Sherman & Wyllie-Echeverria, 2007*), tidal range (*Marsden, 1994*), and interspecific competition for food, space and oxygen with other sessile invertebrates (*Connell, 1961*). Aggregative settlement is thought to improve the probability of survival, increasing the likelihood of finding a suitable settlement site for successful growth and reproduction and mitigating abiotic stress such as wave action and desiccation (*Bianchi & Morri, 1996*; *Thomas, 1996*). In natural conditions, competent larvae of *C. savignyi* tend to sink or swim downwards and become strongly photonegative, displaying a preference for dark or shaded surfaces in areas with reduced water movement and light intensity (*Gulliksen, 1972*; *Schmidt & Warner, 1984*; *Carver, Mallet & Vercaemer, 2006*; *Rudolf et al., 2019*). For this reason, invasive tunicates occur commonly on artificial structures such as floating docks, pontoons, and aquaculture facilities (*Smith et al., 2012*; *Cordell, Levy & Toft, 2013*). Interestingly, our observations of the settlement strategy of *C. savignyi*' larvae during the second bioassay displayed a similar recruitment mechanism, colonizing the edge of the substrates to avoid light exposure. Habitat selection during settlement for sessile benthic invertebrates such as oysters, polychaetes and ascidians is of particular significance because there is no possibility of relocation once the metamorphosis occurs onto a substrate (*Tamburri et al., 2008*). In contrast, *M. galloprovincialis*' larvae can settle and relocate to find an alternative and potentially more appropriate substratum (*Yang et al., 2007*, *Carl et al., 2011*). This phenomenon is termed 'secondary settlement'.

Previous studies also reported that, in natural conditions, *C. gigas*, *S. cariniferus*, *M. galloprovincialis* and *C. savignyi* larvae respond to a wide range of chemical cues that may provide information to secure an appropriate substrate for their post-settlement growth and survival (*Steinberg, De Nys & Kjelleberg, 2002*; *Sánchez-Lazo & Martínez-Pita, 2012*; *Wolf, 2020*). In fact, larvae of these four species are characterized by a gregarious settlement mechanism whereby pediveliger larvae choose to settle in response to the presence of adults, juveniles, or recent recruits of the same species (*Tsukamoto et al., 1999*; *Vasquez et al., 2013*; *Wolf, 2020*; *Montes et al., 2021*). In a field-based study, *Wolf (2020)* investigated the recruitment strategy of the blue tubeworm *S. cariniferus* in the absence of conspecifics. To understand the mechanisms underlying the settlement preferences of *S. cariniferus* in the field, *Wolf (2020)* discusses the "founder and aggregator hypothesis" coined by *Toonen & Pawlik (1994)*, speculating that aggregations must initially develop from a two-step process: solitary larvae first colonize an uninhabited substratum in response to biofilm cues, then gregarious settlement occurs on or near these 'founders'. Based on the results of this study, and since no conspecifics were used to investigate, or induce larval recruitment on the substrates, we postulate that a similar recruitment pattern took place. First, the week-old biofilm layer developed at the polymers' surface attracted the larvae through biological mediation; second, the larvae started to settle in number, favoring specific locations for larval aggregation on the substrates. However, larvae in our experiments were contained in a small volume of water and therefore had limited capacity to select among different settlement sites. Settlement likelihood is greater in this scenario than may be expected in the wild, and the settlement rates reported here should be considered as relative (*i.e.*, relative settlement preference rather than absolute settlement rates). This is particularly the case for *C. savignyi*, which has lecithotrophic (non-feeding) larvae that have limited capacity to extend their free-swimming duration (*Cahill et al., 2016*). It is likewise important to note that *S. caraniferus* was chemically induced to settle, and this will have increased absolute settlement rates relative to what might be expected in the wild.

Unlike *C. gigas* and *M. galloprovincialis*, a clear preference for Nylon was observed in both *C. savignyi* and *S. cariniferus* bioassays. The latter species also displayed a preference for PLA. In a previous study, *Cahill et al. (2016)* detected no difference in *C. savignyi* settlement rates for polystyrene, or acrylic substrates. The authors discussed that the apparent insensitivity to surface characteristics might contribute to *C. savignyi*'s invasiveness, with larvae settling on a wide range of available substrates. Our findings against different substrates provide new information that suggests settlement may be elevated for some manmade substrates (*i.e.*, nylon), although noting that settlement did occur on all substrate types we tested. Although no study has yet reported *S. cariniferus* attached to anthropogenic substrates, *Rech, Borrell Pichs & García-Vazquez (2018)* observed several polychaetes species rafting on marine plastic debris in the Bay of Biscay (Spain, Atlantic Ocean), including the congeners *Spirobranchus triqueter*, *Spirobranchus taeniatus*, and other *Spirobranchus* species. These polychaetes were mostly detected on hard plastics, although no polymer characterization was conducted (*Rech, Borrell Pichs & García-Vazquez, 2018*).

More broadly, colonization of marine invertebrates on hard surfaces depends on many substrate features such as physical properties, chemical composition, surface roughness and mechanical attributes (*Brzozowska et al., 2017*; *Siddik et al., 2019*). For example, surface roughness has been reported as one of the major influencing factors determining the recruitment of sessile larvae on hard substrates (*Köhler, Hansen & Wahl, 1999*). Other studies reported that invertebrate larvae can sense surface energy, and adapt to select an optimal substrate (*Rittschof et al., 1998*). Briefly, a high surface energy (HSE) polymer means a strong molecular attraction (*i.e.*, hydrophobic surface), whereas low surface energy (LSE) polymer means a weak molecular attraction (*i.e.*, hydrophilic surface). For example, *Rittschof & Costlow (1989)* and *Gerhart et al. (1992)* demonstrated that *in vitro* larval behavior and settlement strategy of barnacles, bryozoans, and oysters were altered by exposure to surfaces with different energies. The authors reported that barnacles preferred to settle on HSE surfaces, whereas bryozoans, ascidians and oysters seemed to be attracted by LSE surfaces. In this study, LLDPE was the only LSE polymer, whereas PLA, Nylon and Glass were HSE substrates. Our observations of larval recruitment for *C. savignyi*, *S. cariniferus* and *M. galloprovincialis* revealed minimum larval counts on LLDPE, suggesting a preference of these three species for HSE substrates, although statistical differences were calculated only for *C. savignyi* and *S. cariniferus*. This pattern can be explained by the tested substrates' specific physical and chemical properties. For instance, LLDPE is an inert material with limited chemical functionalities (*i.e.* apolar surface), making it difficult for lifeforms to adhere to it (*Abdul-Kader et al., 2009*). In contrast with LLDPE, PLA has chemical functionality (ester groups, C-O-C(=O)-C) that can be easily cleaved by reaction with seawater (*Elsawy et al., 2017*). The ester groups might serve as an energy source for the invasive' larvae itself, or the microbial biofilm they prey on. For nylon, it is the amide group (C-NH-C(=O)-C), which could potentially facilitate biological interactions (*Sudhakar et al., 2007*).

Another factor that might considerably influence larval recruitment onto a substrate is the biological cues emitted from microbial biofilms that develop on most underwater surfaces. Microbial biofilms have long been recognized as an inducer for the settlement of marine invertebrate larvae (*Johnson et al., 1997*; *Hadfield, 2011*). In a pioneer study, *Johnson et al. (1997)* predicted that interactions between marine invertebrate larvae, microbial biofilms and substrate are widespread in the natural system, mainly because biofilms are likely to be encountered in every marine ecosystem. In addition, *Hadfield (2011)* discusses the fact that bacteria may simply signal the presence of a substratum that has been submerged in the sea long enough to accumulate a substantial biofilm and thus, indicate a food source and/or a nontoxic surface for larval recruitment (*Unabia & Hadfield, 1999*). For example, *Satuito, Shimizu & Fusetani (1997)* compared the settlement response of competent pediveliger of *M. galloprovincialis* on surfaces with and without microbial biofilm, highlighting that recruitment was induced within 48 h. In contrast, no settlement was observed during 72 h of experimental exposure on biofilm-free surfaces. Similar observations were reported for bivalves (*Zhao, Zhang & Qian, 2003*), bryozoans (*Dahms, Dobretsov & Qian, 2004*), ascidians (*Wieczorek & Todd, 1997*), and tubeworms (*Shikuma & Hadfield, 2005*), with a correlation between biofilm age and recruitment

success. In this study, the same seawater was used to run the four consecutive bioassays. Considering that bacterial assemblages are highly dynamic in seawater, the bacterial community composition progressively evolved across the experiment, and the overall trends in our data likely reflect bacterial community succession in the recirculating seawater systems as a whole. Because of this experimental artifact and since the bacterial assemblages were similar across substrates, we cannot draw conclusions on the influence of specific bacterial taxa on larval attachment.

An additional aspect that could have influenced our results is the level of chemosensitivity of marine invertebrates, and how plastic leachates can influence chemosensory perception and communication in the marine realm. The ability of sessile marine invertebrates to accurately detect and respond to environmental cues is essential for successful recruitment (*Lecchini et al., 2005*), finding food (*Tomba, Keller & Moore, 2001*), escaping predation (*Kats & Dill, 1998*), and regulating population dynamics and community structure (*Ashur, Johnston & Dixson, 2017*). Although the mechanisms of chemosensory perception for marine invertebrates have long been acknowledged (*Jensen, 1992*), new studies highlight the impact of plastic leachates on larval behaviour (*Silva et al., 2016*). For instance, *Li et al. (2016)* recently demonstrated a significant inhibition of *Amphibalanus amphitrite*' larvae recruitment on glass when exposed to several different polymer leachates (polyvinyl chloride–PVC; polyethylene–PE; and polycarbonate–PC). Moreover. some plastic leachates, notably plasticizers such as phthalates, are cytotoxic and could have influenced larval fitness (*Staples et al., 1997*). However, since the different plastic polymers were exposed for a short period (*i.e.*, 1 week) in the current investigation, we can only speculate that polymer leachates may have played some role in NIS larval recruitment, but additional research is required to test this hypothesis.

If we extrapolate these results to what's occurring in natural conditions, we can argue that the combination of these factors is amplified. For instance, ecological competition for food and space is fierce in the marine realm, particularly for sessile macro-invertebrates. While this study investigated recruitment patterns through specie-specific assays, we can hypothesize that if the four species were combined into a single assay, the results would have been different. In addition, if the biological cues emitted from microbial biofilms induce macro-invertebrates' recruitment, then the diversity of microbial assemblages found in natural conditions would also significantly affect the results. Hence, more research is needed to investigate how these parameters and their potential interactions influence NIS recruitment, particularly in natural conditions.

## CONCLUSION

This study investigated the effect of polymer type on larval recruitment of four notorious invaders under controlled conditions controlled conditions. Understanding the mechanisms involved in recruitment and subsequent transport/spread of NIS on MPD is paramount to address knowledge gaps around biosecurity risks associated with MPD. Investigating the succession of plastisphere communities from micro- to macro-organisms is a critical first step to understand their ecological impact, fate in marine settings, and their capacity to recruit and carry invasive species within or across broad geographic regions

(*Audrézet et al., 2020*). Altogether, results from this study are exciting. They highlight the complexity of interactions between NIS larvae and plastic polymers. Although this investigation was conducted in controlled conditions, we can conclude that several factors and their potential interactions may have influenced the results presented here, including: (i) species-specific larval biological traits and ecology; (ii) physical and chemical composition of the substrates; (iii) biological and chemical cues emitted from the bacterial biofilm and the level of chemosensitivity of the different NIS larvae. Given the persistence and ubiquity of plastic debris in marine settings, MPD will continue to persist and adversely impact our ecological health for decades. Therefore, more research efforts are needed to understand the mechanisms involved in the recruitment of marine pests, and to answer the many knowledge gaps around the biosecurity risks and ecological fate of MPD in marine habitats.

## ACKNOWLEDGEMENTS

We are particularly grateful to Regis Risani and Beatrix Theobald (Scion) for the design and production of the polymers. Robert Abbel (Scion) and Lloyd Donaldson (Scion) for their expertise in polymer chemistry and valuable discussion; Hannah Mae and Julien Vignier for providing competent oyster larvae; Devin Golub and Juliette Butler for their help with the larval cultures; Tim Dodgshun for his help in preparing the 'wetlab' to run the experiment; Olga Panto and Grant Northcott for valuable discussions.

### Funding

This study was supported by the Aotearoa Impacts and Mitigation of Microplastics (AIM2) and The Marine Biosecurity Toolbox projects (Ministry of Business, Innovation, and Employment, New Zealand, Endeavour Fund C03X1802 and CAWX1904). The funders had no role in study design, data collection and analysis, decision to publish, or preparation of the manuscript.

### Grant Disclosures

The following grant information was disclosed by the authors:
Ministry of Business, Innovation, and Employment, New Zealand, Endeavour Fund: C03X1802 and CAWX1904.

### Competing Interests

Susanna Wood, Anastasija Zaiko and Xavier Pochon are Academic Editors for PeerJ.

### Author Contributions

- François Audrézet conceived and designed the experiments, performed the experiments, analyzed the data, prepared figures and/or tables, and approved the final draft.
- Anastasija Zaiko conceived and designed the experiments, analyzed the data, authored or reviewed drafts of the article, and approved the final draft.

- Patrick Cahill performed the experiments, authored or reviewed drafts of the article, and approved the final draft.
- Olivier Champeau performed the experiments, authored or reviewed drafts of the article, and approved the final draft.
- Louis A. Tremblay conceived and designed the experiments, authored or reviewed drafts of the article, and approved the final draft.
- Dawn Smith conceived and designed the experiments, authored or reviewed drafts of the article, and approved the final draft.
- Susanna A. Wood conceived and designed the experiments, authored or reviewed drafts of the article, and approved the final draft.
- Gavin Lear conceived and designed the experiments, authored or reviewed drafts of the article, and approved the final draft.
- Xavier Pochon conceived and designed the experiments, analyzed the data, authored or reviewed drafts of the article, and approved the final draft.

## DNA Deposition

The following information was supplied regarding the deposition of DNA sequences:
The raw sequence reads are available at NCBI: PRJNA836386.

## Data Availability

The raw sequence reads are available at NCBI: PRJNA836386.
https://www.ncbi.nlm.nih.gov/sra/PRJNA836386.

## Supplemental Information

Supplemental information for this article can be found online at http://dx.doi.org/10.7717/peerj.14549#supplemental-information.

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
