# Peer review of "Does plastic type matter? Insights into non-indigenous marine larvae recruitment under controlled conditions"

_PeerJ, doi:10.7717/peerj.14549_

## Round 0.1 · original submission · Major Revisions

Both reviewers agreed that this is a valuable contribution to understanding the impact of plastic substrate type on larval recruitment and potential for transport.

However, both reviewers provide detailed suggestions that will greatly improve clarity throughout.

Introduction: Carefully consider the references cited. Many more current suggestions were provided.

Methods: Clarity is needed on the experimental design, particularly with updates to Figure 1.

Results: Reviewer 2 offers detailed feedback on the presentation and analysis of the microbial community results.

Discussion: While the Discussion walks through each of the major results, I agree with Reviewer 1 that the section could benefit from attention to the implications/application of these results in the natural environment.

I also suggest careful proof-reading after revisions are complete and prior to submission. Lines 239-242 are repetitive.

·

Basic reporting

No comment - the study was very well reported.

Experimental design

No comment

Validity of the findings

No comment

Additional comments

Your paper is really well done, I think that your findings will be a really great addition to the body of literature. I did not have any major comments / concerns about the experimental design, interpretation of results, or conclusions. I did have some minor comments as I read through, which I have written out below. Overall, I think this is a great paper and exciting study. Congratulations!

Introduction
Very effective introduction! Great job.
52-53: Great paragraph! Just took issue with the last sentence (maybe because I’m a freshwater MP scientist!), but there is also lots of plastic pollution in freshwater and terrestrial, arguably a higher load given the relative volume of ocean to lake.
56: I would argue that “release and adsorption of associated pollutants” isn’t an impact of plastic debris – it’s a process, which can have biological impacts like endocrine disruption. For that matter, “transfer of micro-particles through the planktonic food web” isn’t an impact of plastic pollution either. I don’t have issues with the phrases themselves, more with the structure of the paragraph. Since you start off by talking about impacts to marine biota, you should use actual impacts as your examples.
68-70: Great ending to this paragraph!
78: Nit-picky, but (LSE) should go before polymer in this sentence. Can you also comment on why the surface energy should matter in this study? I’m not familiar with this term, so it would be helpful to make the relationship between colonizing and surface energy more clear. … Now that I’ve read the discussion, I see that you do make the relationship clear. But I still think explaining it very briefly in this section would be helpful.

Methods
137: Did you choose these additives, or is this a description of what additives are already associated with the plastics you purchased? Maybe add a sentence at the end of this paragraph to contextualize these additives – e.g. they are commonly added to these polymer types
148: I was confused about the explanation for the settlement assay, until I read the figure 2 caption. The part I didn’t understand was why you did 7 days and then added the larvae and waited 2 days – so maybe just say “…for 7 days to allow biofilm growth”. Or something like you have in Line 155, if you moved that up a bit earlier I think it would help the reader understand the 7 / 2 day timeline.
164: How was the location of the larvae on the token assessed?
219: Missing a word in this sentence – Tukey’s honestly significant difference test

Results
258-260: I was unsure what you meant by “lower/upper part of the tested substrate” until I saw the figure. Maybe you can add “upper part of the tested substrate (nearer to water surface)” or something to clarify what you mean.

Discussion / Conclusion
These sections were also well done. One thing I felt was missing was the implications of the results in the environment. In the conclusion you state that this type of work helps to understand ecological impact, fate, capacity to carry invasive species… I guess I was hoping you would comment on how your specific results inform our understanding of ecological impact / consequences of MP colonization.

Figures
Figure 1: On the first panel, where you have the two schematics, can you add some text to help orient the diagrams? Like “birds eye view” or “side view”… something like that to explain where the diagram is in relation to the picture in the middle panel. If you don’t want to add more text, you could add it to the figure caption, and use figure labels, A, B, C….

Reviewer 2 ·

Excellent Review

This review has been rated excellent by staff (in the top 15% of reviews)
EDITOR COMMENT
This is indeed an excellent review. Thank you! The careful reading of the manuscript was combined with detailed and constructive comments for the authors. The review provides a clear path forward for revision.

Basic reporting

Overall, this manuscript was clear, well-written and interesting and I enjoyed reading it. I do have several comments for this section that I believe will improve the clarity and rigour of the manuscript:
1. A few references in the introduction could do with being updated/supplemented with more recent numbers/citations:
- Plastics Europe - there is a 2021 report that the 2018 version should be replaced with (line 47)
- Jambeck et al. 2015: there are several newer studies from the same group that provide additional information, e.g. https://www.science.org/doi/10.1126/sciadv.abd0288, https://www.science.org/doi/10.1126/science.aba3656 (like 53)
- While Dussud et al. (line 66) do report on "putative pathogens", these are reported based on taxonomy only and I would encourage the authors to replace this with a report that actually tests pathogenicity/virulence.
- There are other studies by Kirstein et al. (line 66) that look at pathogens, but the 2018 study isn't one of them.
- While both Gambarini et al. 2021 and Lear et al. 2021 do examine plastic degraders, neither of them report on plastisphere communities directly and I'd therefore encourage the authors to also cite studies that are actual characterisations of the plastisphere for this statement.

2. The authors state that little is known about the plastisphere involved in community succession (line 69) - and I don't disagree - but there are now quite a few studies that do look at community succession (e.g. https://www.frontiersin.org/articles/10.3389/fmars.2022.841142/full, https://microbiomejournal.biomedcentral.com/articles/10.1186/s40168-021-01054-5, https://pubmed.ncbi.nlm.nih.gov/35537304/, https://link.springer.com/article/10.1007/s00248-019-01424-5). Please also see further comments below regarding whether the study actually addresses community succession or not.

3. Line 79: please add a citation for the low/high surface energy, and perhaps also the descriptor of hydrophobic/hydrophillic, as is given in the discussion.

4. Figures:
- Figure 1: I found it difficult to work out where the four panels fit with the chamber shown in the centre. Is the bottom left diagram a top down view? If so, this would be useful to show. Given this, it was unclear to me whether settlement was assessed on both sides of the substrate (or whether they were placed on top of something else?) or only on the top. Some additional explanation is needed to help readers orient the diagrams.
- Figures 2 and 3: Note the missing 's' on the end of 'specie' in both figures. Also on Figure 2 please clarify whether the diagrams of the panels are top-down views, and add 16S rRNA 'gene' to Figure 3.
- Figure 4: I was unclear on how/whether this was quantified, or if it was just through observations? Some microscope images would really help to strengthen this.
- Figure 5: "Significant p-values are highlighted in bold" - it seems like only the significant p-values are shown? Please clarify. Also note "data extreme points": "extreme data points".
- Figure 6: It would be useful to add the species of larvae to this, as well as the assay number. I would also suggest that the authors consider switching the stacked bar charts for something more quantitative (like a heatmap with numbers written into the boxes, for example). As it stands, it is very difficult to verify what the authors say about abundance, and the colours chosen would also not be good from an accessibility standpoint (particularly for anyone with colour-blindness). It was also unclear what is actually represented here - I struggle to believe that the top 10 families actually account for 100% of all reads at all time points/in all substrates. See further comments on this below.

5. I appreciate that the authors have shared the raw sequencing data, but it would be beneficial if they could also share the processed ASV tables that were used for microbial community analyses, as well as raw count tables for the larval settlement data (showing data for each replicate). If images from the microscopy of the settled larvae also exist then this would be useful to add, too.

Experimental design

This study is interesting and I believe that the question being addressed is one of importance, and the methods used are appropriate. However, I feel that the microbial community analysis is currently a bit lacking and doesn't seem to me like it is able to address microbial community succession, as claimed by the authors (line 71). See my specific comments below:
1. The authors claim to investigate microbial community succession (line 71), and while they touch on this in the discussion, their methods and results sections don't make it clear that they are doing this through the different assays. Unless I have misunderstood, the authors use the same seawater for all assays, but use different plastic materials for each assay. Therefore, the microbial communities on the plastic materials are all only characterised at a single time point and the authors therefore cannot say that they characterise community succession. It was also not clear to me whether the seawater that was used for all assays all came from the same pool, or was kept the same within each tank. The PERMANOVA should presumably also account for the different starting microbial communities.

2. Regarding the microbial community analysis, it was quite unclear to me what was actually shown. If only the top 10 families were shown, then these should be shown as their abundance within samples, not just relative to each other. It would also have been useful to visualise the communities on PCoA plots (or similar), possibly both for all assays and for each assay individually - using a metric that also accounts for phylogeny, like Weighted/Unweighted UniFrac, would also be beneficial. Furthermore, the authors should carry out differential abundance analysis (e.g. https://www.nature.com/articles/s41467-022-28034-z, https://microbiomejournal.biomedcentral.com/articles/10.1186/s40168-017-0237-y) in order to support the results that they present on comparisons between microbial families.

Minor comments:
- Line 160: I think that it would probably be preferable to keep the tokens used for bacterial characterisation in the experimental conditions for longer, rather than keeping them on ice as I would expect keeping them on ice to have more impact on the community than an additional 2 hours (on top of 9 days) in experimental conditions.
- Line 221: 16S rRNA gene amplicon sequencing is not metagenomics. Please edit this appropriately.
- Line 226-227: It would be clearer for the reader if this read as more like "Sequences were trimmed at a length of 220 for both forward and reverse reads, 2 or 6 errors were allowed for forward or reverse reads respectively"... etc. and the raw code made available as supplementary data.
- Was any further filtering done other than removal of singleton ASVs? And was this on a per sample or per dataset basis?
- Lines 234-236: More details on the removal of ASVs are needed here. Was a random number of ASVs just removed from samples? Personally, I would prefer to not see any ASVs be removed from samples, but rather see that any controls group very separately from samples on an ordination plot, or similar.
- Line 300: How was homogeneity tests?

Validity of the findings

I have nothing further to add in this section, although may have more after the additional details suggested above have been provided.

I did enjoy reading the discussion and felt that this was generally well supported by the findings, I just felt that those parts relating mainly to the microbial community analysis require further work.

---

## Round 0.2 · Minor Revisions

The manuscript is much improved following the major revision.

I agree with additional comments provided by the reviewer. Attention should be given to the presentation of microbial community results (Figure 6 and associated text). To this end, I think it is bold to include statements in the Discussion (beginning) and Conclusion that imply that "biological and chemical cues emitted from the bacterial biofilm" influence settlement of the larvae tested. The bacterial microfilm was consistent across replicates, but evolved across assays with different larval species. You have no evidence of bacterial community influencing recruitment, even if it does ultimately play a role. I do agree that a more interesting and relavent second hypothesis would be focused on the role of the bacterial community in recruitment. This would also then be parallel with the first hypothesis.

Two minor copy-editing comments:
Line 62 - check noun-verb agreement
Line 275 - needs editing - remove "different"

I look forward to receiving the updated draft with these minor revisions.

Reviewer 2 ·

Basic reporting

First of all, I apologise for my slow review delaying the manuscript. I appreciate the changes that the authors have made and believe that the manuscript is now much improved, however, there were a few points that I made previously where I think that my point has been misunderstood. Hopefully I have clarified that now.
- Figure 6: My point here was not that raw read counts should be used instead, rather that the y-label states that what is shown is the “proportion of the community”, but I think that this should be “proportion within top 10 families”, or similar. Please correct me if I’m wrong, but I would be very surprised if the axis should sum to 1 for only 10 families. The authors’ response also failed to address my comment about accessibility, as well as that a stacked bar chart is not a quantitative way of looking at the data. For example, without the processed ASV tables – as clarified below – there is no way for me to verify the abundance numbers given in lines 297-301.
- When I mentioned “processed ASV tables”, what I meant was a table that showed ASVs as rows and samples as columns, along with the number of reads for each ASV in each sample to fill in the table. This table would also show the taxonomic classifications for all ASVs. Also, in the raw settlement data it would be useful to indicate the microcosm that each substrate came from (I assume that this is the ordering of sample but specifying this exactly would be useful).
- PCoA plot: I personally feel that it would be useful to have a similar plot as supplementary information to back up what is in the text regarding the PERMANOVA (although using either Bray-Curtis for both or weighted unifrac for both), although I will leave this to the authors/editor. However, I assume that this ordination was performed for all assays together and then each was plotted separately? What is included on the same plot/panel should be calculated together.

Additional comment:
The first hypothesis is addressed directly in the discussion while the second isn’t. Given that there aren’t any large differences between the substrates in terms of the community composition I think that it would be fine just to include something short that specifically addresses this. However, I think that given the focus of the discussion around microbial communities being largely being centered on how they drive larval recruitment I think it might make more sense to reframe the second hypothesis to include that differences in the colonising biofilm would drive differences in larval recruitment among the substrates, or something along these lines.

Experimental design

Figure 4: The reason for not including microscope images is perfectly acceptable to me, although it would be useful to note something to this effect in the figure legend – for any reader that may have the same questions as me.
The seawater: It would be useful to add some of the detail provided to me in the text, just something to the effect of “seawater was not replaced between assays” would be fine.
Line 310 homogeneity: please add the details on homogeneity testing to the manuscript.

Validity of the findings

Please see comments above.

Additional comments

Overall I really enjoyed reading the manuscript and think that it's a really valuable contribution to the literature.

---

## Round 0.3 · accepted · Accept

I am pleased to accept this manuscript for publication. It is clear that you have addressed all of the reviewers' comments. Once available, carefully review the proofing PDF, as minor spacing or punctuation errors exist in the latest version of the reviewing PDF.